# Significant Enhancement of the Capacity and Cycling Stability of Lithium-Rich Manganese-Based Layered Cathode Materials via Molybdenum Surface Modification

**DOI:** 10.3390/molecules27072100

**Published:** 2022-03-24

**Authors:** Yijia Shao, Zhiyuan Lu, Luoqian Li, Yanni Liu, Lijun Yang, Ting Shu, Xiuhua Li, Shijun Liao

**Affiliations:** The Key Laboratory of Fuel Cell Technology of Guangdong Province, School of Chemistry and Chemical Engineering, South China University of Technology, Guangzhou 510641, China; ceyjshao@mail.scut.edu.cn (Y.S.); 15625159545@163.com (Z.L.); 202020123782@mail.scut.edu.cn (L.L.); liuyanniscut@163.com (Y.L.); yanglijun028@gmail.com (L.Y.); shuting6418@163.com (T.S.); lixiuhua@scut.edu.cn (X.L.)

**Keywords:** lithium-rich cathode, Mo-based surface modification, lithium ion batteries

## Abstract

Lithium-rich manganese-based layered cathode materials are considered to be one of the best options for next-generation lithium-ion batteries, owing to their ultra-high specific capacity (>250 mAh·g^−1^) and platform voltage. However, their poor cycling stability, caused by the release of lattice oxygen as well as the electrode/electrolyte side reactions accompanying complex phase transformation, makes it difficult to use this material in practical applications. In this work, we suggest a molybdenum surface modification strategy to improve the electrochemical performance of Li_1.2_Mn_0.54_Ni_0.13_Co_0.13_O_2_. The Mo-modified Li_1.2_Mn_0.54_Ni_0.13_Co_0.13_O_2_ material exhibits an enhanced discharge specific capacity of up to 290.5 mAh·g^−1^ (20 mA·g^−1^) and a capacity retention rate of 82% (300 cycles at 200 mA·g^−1^), compared with 261.2 mAh·g^−1^ and a 70% retention rate for the material without Mo modification. The significantly enhanced performance of the modified material can be ascribed to the formation of a Mo-compound-involved nanolayer on the surface of the materials, which effectively lessens the electrolyte corrosion of the cathode, as well as the activation of Mo^6+^ towards Ni^2+^/Ni^4+^ redox couples and the pre-activation of a Mo compound. This study offers a facile and effective strategy to address the poor cyclability of lithium-rich manganese-based layered cathode materials.

## 1. Introduction

With the development of renewable energy technology and electric vehicles, conventional cathode materials, such as LiCoO_2_, LiMn_2_O_4_, LiFePO_4_, and Li(Ni_x_Mn_y_Co_1−x−y_)O_2_, are unable to meet the requirements of next-generation lithium ion batteries because they cannot achieve the high capacity, high energy density, and high cyclability demanded for electric vehicles. Thus, new cathode materials with a high energy density are urgently needed [1,2,3]. Lithium-rich manganese-based cathode materials (LMR), specifically low-Co and -Ni content lithium-rich materials, such as 0.5 Li_2_MnO_3_ 0.5 LiNi_0.33_Co_0.33_Mn_0.33_O_2_ or Li_1.2_Mn_0.54_Ni_0.13_Co_0.13_O_2_, have been recognized as some of the most promising cathode candidates because of their ultra-high specific capacity (>250 mAh·g^−1^) and power density (>1000 Wh g^−1^) and low cost. It has been demonstrated that the reduction of the lattice oxygen of the Li_2_MnO_3_ component under a high charge voltage of 4.8 V (vs. Li^+^/Li) plays a crucial role for the ultra-high discharge capacity of LMR. However, due to the lower electronic conductivity of Li_2_MnO_3_, its oxygen releasing and phase transition from layer to spinel, as well as the degradation of electrolytes, LMR suffers from some crucial drawbacks, such as its poor cycling stability, low rate capability, and voltage fading, hindering its practice application [4,5,6,7,8].

To address the above stated problems, many efforts have been devoted to improving the performance of the materials, such as morphology control [9,10,11,12,13,14,15], surface modification [16,17,18,19,20,21,22,23], heteroelements doping [24,25,26,27,28,29,30,31,32], etc. Among them, newly emerged surface modification is regarded as the most effective way to mitigate phase transition triggered by oxygen release and electrolytes etching. Furthermore, surface modification is also an effective way to enhance the Li^+^ and electronic conductivity of LMR materials. It is confirmed by our group’s previous work [33] that the doping of the molybdenum (Mo) element in the LMR material can retard the Jahn–Teller effect of Mn and effectively impede the phase transformation caused by the release of lattice oxygen. Meanwhile, it is found that the doping of Mo in layer-structured LMR material could expand the interlayer spacing and consequently accelerate the bulk diffusion of Li^+^ [34]. Furthermore, coating LMR materials with molybdenum oxide or molybdenite could improve the interface conductivity of Li^+^ and electrons effectively. 

Inspired by the previous research works and results, we attempted to improve the performance of LMR materials through surface modification. It is found that the discharge capacity and cycling stability of Li_1.2_Mn_0.54_Ni_0.13_Co_0.13_O_2_ could be enhanced significantly through modification of the material with molybdenum by using the ammonium molybdate (H_8_MoN_2_O_4_) as a modifier (Figure 1). Based on the characterization results, a nanometer layer containing the Mo compound is formed on the LMR surface, and some of the Mo^6+^ ions penetrate the material’s crystal lattice. The Mo-modified LMR material exhibits a significantly enhanced electrochemical performance, and especially a considerably enhanced electrochemical rate performance and thermodynamic stability, giving the material outstanding cycling stability.

## 2. Results and Discussion

Figure 1 shows the XRD patterns of LMR (denotes Li_1.2_Mn_0.54_Ni_0.13_Co_0.13_O_2_) and LMR-Mo (denotes Mo-surface-modified Li_1.2_Mn_0.54_Ni_0.13_Co_0.13_O_2_). For the sample LMR, all of the dominant diffraction peaks and apparent splitting peaks of 006/102 and 018/110 of both materials can be indexed to the R-3m space group of the hexagonal α-NaFeO_2_-type layered structure, without an obvious impurity phase [35,36]. The weak superstructure reflections of both materials between 20° and 25° indicate the existence of the Li_2_MnO_3_ phase, which is the typical feature of lithium-rich cathode materials, arising from the superlattice ordering of Li and Mn in the transition metal layers of the Li_2_MnO_3_ phase [37,38,39,40].

After the modification process by H_8_MoN_2_O_4_, the (003) peak of LMR-Mo was shifted to a lower angle, compared with the LMR sample. This should be caused by the larger ionic radius of Mo^6+^ (0.59 Å) than Mn^4+^ (0.53 Å), indicating that part of the Mo atoms/ions are inserted into the LMR lattice. The lattice parameters of LMR-Mo, calculated by Jade software and shown in Table 1, also provide evidence that the lattice was expended from a and c directions. This lattice expansion is beneficial to the diffusion of Li^+^ in the materials’ lattice, resulting in the enhancement of the performance of the materials. 

As shown in Table 1, the ratio of I_(003)_/I_(104)_ of LMR-Mo is larger than that of the LMR sample, indicating the decrease in Ni^2+^ entering the Li^+^ layer. In other words, the Mo surface modification effectively restrained the cationic mixing. This will result in a better electrochemical performance, according to the literature [41,42]. Further, obvious changes can be observed in the weak reflections from 20° to 25° after modification: LMR only exhibited two broad peaks, whereas LMR-Mo exhibited four sharp peaks, because the bonds between Mo and O were stronger than those between Mn and O in Li_2_MnO_3_ [43]. As Mo atoms were doped into the Li/Mn layer, they would activate and stabilize the local environment of the O atoms in Li_2_MnO_3_.

SEM images of LMR and LMR-Mo are presented in Figure 2a and Appendix A. Both materials show an assembled morphology of pebble-like particles. TEM and high-resolution TEM images of LMR and LMR-Mo are illustrated in Figure 2b and Appendix A. It was clear that an approximate 1 nm-thick, rough nanostructure layer was uniformly coated on the particle surface, confirming the successful surface modification/coating of Mo, and that clear lattice fringes (see insert of Figure 2b) and electron diffraction (ED) patterns (Figure 2c) corresponded to the (003), (012), and (006) plane of the LiTMO_2_ (here, TM denotes transition metal element), and (020), (111), (−113) plane of the Li_2_MnO_3_ component, respectively [44,45]. These results are consistent with the results of XRD. Furthermoe, the EDS mapping results (Figure 2d) further confirm the successful preparation of Mo-modified lithium-rich cathode materials and the surface modification of Mo on LRM materials.

To investigate the changes in chemical states brought about by Mo modification, XPS analysis was conducted in this work. Figure 3 shows the spectra of Mn, Ni, and Mo in LMR and LMR-Mo. For the LMR sample, the binding energies of Mn 2p3/2 and Mn 2p1/2 were 642.5 eV and 654 eV, corresponding to the Mn^4+^ of MnO_2_ [46,47]. In comparison, the binding energies of Mn 2p3/2 and Mn 2p1/2 for the LMR-Mo sample were slightly down-shifted, indicating that the charge transfer to Mn from Mo occurred after surface modification with Mo. As shown in Figure 3b, all of the Ni 2p3/2 were divided into two peaks for Ni^2+^ and Ni^3+^; after Mo modification, the Ni^2+^ content increased from 27% for LMR to 41% for LMR-Mo. Evidently, Mo surface modification led to an increase in Ni^2+^ and a decrease in Ni^3+^, improving the Ni^2+^/Ni^4+^ ratio and thus boosting the cathode material’s specific capacity. Figure 3d shows the XPS spectrum of Mo. The binding energy of Mo 3d5/2 was around 232.5 eV, indicating that Mo existed in the material in the +6 oxidation state, implying that the coating layer may be composed of molybdenum oxide or molybdenate.

Figure 4a shows the initial charge/discharge profiles of batteries with LMR or LMR-Mo as cathodes, respectively. Generally, the charging process of lithium-rich manganese-based layered cathode materials can be divided into two stages [48,49,50,51]. In the first stage (potential < 4.45 V), the Li+ are released and charge compensation is achieved by transition metal ion oxidation in LiTMO_2_, specifically Ni^2+^/Ni^4+^ and Co^3+^/Co^4+^; the Li_2_MnO_3_ phase does not participate in the reaction and only plays the role of stabilizing the material’s structure. In the second stage, when the voltage is higher than 4.45 V, Li^+^ is released and charge compensation is achieved by the O^2–^ in Li_2_MnO_3_; at the same time, Li_2_MnO_3_ turns into the MnO_2_ active component. Figure 4a shows that LMR and LMR-Mo had two similar charge/discharge curves and delivered respective discharge capacities of 261.2 and 280.9 mAh·g^–1^ at 0.1 C. Surface modification with Mo thus brought about a 10% increase in the discharge capacity. In fact, the maximum discharge specific capacity of LMR-Mo reached 290.5 mAh·g^–1^ after typical electrochemical activation. LMR-Mo also exhibited a better rate performance (Figure 4b) and better cyclability than LMR. After 300 cycles at 1 C, the capacity of the battery with the LMR-Mo cathode was 181.8 mAh·g^–1^, with the capacity retention being up to 82%, whereas the battery with the LMR cathode had a capacity of only 124.5 mAh·g^–1^, with approximately a 70% capacity retention (Figure 4c). The dQ/dV curves of LMR and LMR-Mo (Appendix A, Appendix A) also illustrated that, even after 300 cycles, the LMR-Mo still had a higher cycle retention. All of the above results indicated that the surface modification with Mo significantly improved the electrochemical performance of LMR. We suggested that the formation of a Mo compound nanolayer on the surface of the material particles may have lessened the electrolyte’s corrosive effect on the active material, resulting in a better long cycling stability.

CV analysis was used to further investigate the effects of Mo surface modification on the redox reactions during the charge/discharge process. Figure 5 shows the CV curves of the initial three cycles for LMR and LMR-Mo in the potential range of 2.0–4.8 V. For the initial charge processes, both samples exhibited two main oxidation peaks, with those of the LMR-Mo sample near 4.111 V and 4.734 V corresponding to the Ni^2+^/^4+^ and oxygen compensation reactions. After modification with Mo, these two redox reactions presented a completely inverse trend as a result of the increased Ni^2+^ content and the stability of the Mo-O bond, which was consistent with the XPS analysis results described earlier and the initial charge curve at 0.1 C. For the first discharge process, there were two reduction peaks around 3.735 V and 3.325 V, corresponding to the redox of Ni^4+^/^2+^ and Mn^4+^/^3+^. All peak potentials are higher than those of the LMR sample, indicating that the Mo modification strategy effectively increased the reversibility of the Ni^2+^/^4+^ redox and hindered the phase transition from layer to spinel. In the next two CV curves, the same trend can be observed, indicating the reduction in irreversible capacity loss, as well as oxygen release, at the activation of Li_2_MnO_3_ [52,53,54,55,56,57], which resulted in a better initial coulombic efficiency and cycling stability for this lithium-rich cathode material.

To further explore the effect of Mo surface modification on the kinetic properties of LMR, an EIS measurement was conducted in the open-circuit state after charging to 4.8 V (vs. Li/Li^+^). As shown in Figure 6, the Nyquist curves of both materials are composed of a semicircle and an inclined straight line. The semicircle in the high-frequency region corresponds to the charge-transfer resistance (R_ct_), which indicates the impedance between the interior and the interface of the active materials [21,58]. Through simulations, we obtained the R_ct_ values of LMR (185.9 Ω) and LMR-Mo (152.1 Ω), showing that, after Mo modification, the charge transfer resistance between the positive electrode material and the electrolyte (R_ct_) was significantly reduced; hence, the kinetics was effectively accelerated, resulting in a higher capacity and rate performance. The slope of the inclined line in the low-frequency region is related to the Warburg impedance resistance (Z_w_), which reflects the ease of the solid-phase diffusion of lithium ions in electrode materials [59,60]. In addition, from the relationships [23,61,62] between Z_re_ and ω^–1/2^ in the low-frequency region, the lithium-ion diffusion coefficients (D_Li_^+^) of all of the samples were estimated to be 3.25 × 10^−17^ and 4.38 × 10^−17^ cm^2^S^−1^ for LMR and LMR-Mo, respectively. Clearly, LMR-Mo demonstrated a higher lithium-ion diffusivity than LMR, which fits with the electrochemical performance results.

## 3. Materials and Methods

### 3.1. Preparation and Characterization

The LMR sample was prepared using a spray-drying process. First, stoichiometric amounts of transition metal acetate (Co(CH_3_COO)_2_·4H_2_O, Mn(CH_3_COO)_2_·4H_2_O, Ni(CH_3_COO)_2_·4H_2_O), lithium hydroxide(LiOH·H_2_O, excess 5%), ammonium hydrogen carbonate, and hexadecyl trimethyl ammonium bromide (as a surfactant) were separately dissolved in deionized water to obtain transition metal acetate solution (solutions A), lithium hydroxide solution (solution B), and ammonium hydrogen carbonate and hexadecyl trimethyl ammonium bromide solution (solution C). Afterward, solutions A and B were dripped slowly into solution C, with vigorous stirring. After aging by 12 h at room temperature, the resulting suspension was spray-dried by a spray-drying instrument (SP-1500, Nanbei Instrument Limited, ShunYi, Beijing, China) to yield a homogenous powder. Finally, the LMR cathode materials were obtained by first heating the powder at 450 °C for 5 h and then calcining it at 900 °C for 12 h in air.

Surface modification of LMR was carried out as follows. First, H_8_MoN_2_O_4_ was dissolved in ethanol to obtain a solution containing Mo, with a molar ratio of Mn/Mo of 17/1. Then the LMR powder was dispersed in the solution for surface adsorption. The solvent was then evaporated via continuous stirring at 70 °C for 8 h, followed by drying at 80 °C for 12 h, and further calcining at 600 °C for 6 h in air.

Phase identification was carried out using an X-ray powder diffractometer (XRD TD3500, Tong-da, Dalian, China), and the morphology and surface state of the samples were examined with a scanning electron microscope (SEM SU8220, Hitachi, Tokyo, Japan), a transmission electron microscope (TEM JEM-2100HR, JEOL, Tokyo, Japan), and an X-ray photoelectron spectroscope (XPS K-Alpha, Thermo-VG Scientific, Waltham, MA, USA).

### 3.2. Electrochemical Measurement

The electrochemical properties of the cathode materials were evaluated with a CR2016 coin-type cell, using a fresh lithium foil as the reference electrode, a polypropylene microporous film (Celgard 2500, Charlotte, NC, USA) as the separator, and 1 mol L^–1^ LiPF6 solution as the electrolyte, obtained by dissolving LiPF6 in a mixed organic solvent of dimethyl carbonate and ethylene carbonate (1:1 in volume ratio). The testing cathode (working electrode) was prepared as follows. First, the prepared active material (80 wt%), acetylene black (10 wt%), and polyvinylidene fluoride (10 wt%) were weighed in proportion. Then, N-methyl pyrrolidone was added dropwise and the mixture was stirred thoroughly. Next, the mixed slurry was uniformly coated on aluminum foils and dried overnight at 80 °C, then cut with a manual slicer to obtain a positive electrode sheet with a diameter of 14 mm. The coin cell was assembled in an argon-filled glove box.

All of the electrochemical tests in this experiment were conducted at room temperature, and the testing voltage window was 2.0–4.8 V (vs. Li/Li^+^). Galvanostatic charge/discharge testing of the prepared coin battery was conducted on an RT-3008w battery testing system (Xingwei, Shenzhen, China). In this work, we specified 1 C as 200 mAh·g^–1^. Cyclic voltammetry (CV) tests were carried out on an Autolab PGSTAT302N electrochemical workstation (Metrohm, Autolab PGSTAT302N, Herisau, Switzerland) at a scanning rate of 0.1 mV·s^–1^. The electrochemical workstation used for electrochemical impedance spectroscopy (EIS) was an IM6e (Zahner, IM6e, Kronach, Germany), and a frequency range of 0.1–100 KHz with a sinusoidal voltage signal 5 mV in amplitude was the perturbation.

## 4. Conclusions

In this work, we intensively investigated the surface modification of the lithium-rich cathode material Li_1.2_Mn_0.54_Ni_0.13_Co_0.13_O_2_ with Mo. After modification, a surface layer of molybdenum oxide or molybdenite involved with a thickness of 1–2 nm was formed on the Li-rich cathode material particles, which effectively improved the surface kinetics and lessened the electrolyte corrosion of the positive material during long cycling, ultimately giving the material a significantly enhanced activity and stability. Based on the characterization results, it is found that the Mo^6+^ simultaneously entered the surface layer of the Li-rich material, enlarging the Li^+^ slab and facilitating the diffusion of Li+ into the layered structure. The surface modification with Mo also increased the ratio of Ni^2+^/Ni^4+^ redox couples and preactivated the Li_2_MnO_3_ component, thereby greatly improving the cathode material’s discharge capacity and cycling stability. Our work offers a modification strategy for enhancing the electrochemical performance of Li-rich materials, and could be beneficial for the commercialization of lithium-rich manganese-based cathode materials.

## Data Availability

Not applicable.

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
