# Peer review of "Significant Enhancement of the Capacity and Cycling Stability of Lithium-Rich Manganese-Based Layered Cathode Materials via Molybdenum Surface Modification"

_molecules, 2022, doi:10.3390/molecules27072100_

Round 1
Reviewer 1 Report
The manuscript is interesting but the discussion of the results can be improved, in particular for what concerns the structural characaterizations.
Many of the authors claims are based on the XRD data of fig.1, but it is very well known that XRD spectra are the integral representation of structural information coming from a large area of the investigated sample, i.e. that one interacting with the X-ray beam.
The authors have reported nice TEM images but not the corresponding ED patterns that would have to be produced in order to confirm (or not) on a nanoscale the authors' hypothesis made on the macroscale on the base of XRD data.
Overall, it is not clear the scientific rationale for which the authors used many and different characterization methods without a strong correlation of the achieved results. For some aspects, the manuscript appears like a data and image mere gallery...
In such a framework, my recommendation is for a major and mandatory revision of the manuscript.
Author Response
Point 1: Many of the authors claims are based on the XRD data of fig.1, but it is very well known that XRD spectra are the integral representation of structural information coming from a large area of the investigated sample, i.e. that one interacting with the X-ray beam. The authors have reported nice TEM images but not the corresponding ED patterns that would have to be produced in order to confirm (or not) on a nanoscale the authors' hypothesis made on the macroscale on the base of XRD data.
Response 1: Thank you very much for your valuable and professional comments. Yes, XRD was frequently used in our work, because it is an efficient extensively used characteristic tool for the investigation of LIB materials [1-5], and it is also an easily accessible tool. Regarding the TEM images, it is really a fault not to take corresponding ED patterns simultaneously. Thanks again for your valuable advice, which will be helpful for us to improve our future works.
- Fang, J.; An, H.; Qin, F.; Wang, H.; Chen, C.; Wang, X.; Li, Y.; Hong, B.; Li, J., Simple Glycerol-Assisted and Morphology-Controllable Solvothermal Synthesis of Lithium-Ion Battery-Layered Li1.2Mn0.54Ni0.13Co0.13O2 Cathode Materials. Acs Appl. Mater. Interfaces 2020, 12, (50), 55926-55935.
- Cui, S.-L.; Gao, M.-Y.; Li, G.-R.; Gao, X.-P., Insights into Li-Rich Mn-Based Cathode Materials with High Capacity: from Dimension to Lattice to Atom. Adv. Energy Mater. 2022, 12, (4), 2003885.
- Ma, Q.; Chen, Z.; Zhong, S.; Meng, J.; Lai, F.; Li, Z.; Cheng, C.; Zhang, L.; Liu, T., Na-substitution induced oxygen vacancy achieving high transition metal capacity in commercial Li-rich cathode. Nano Energy 2021, 81, 105622.
- Zhang, C.; Wei, B.; Jiang, W.; Wang, M.; Hu, W.; Liang, C.; Wang, T.; Chen, L.; Zhang, R.; Wang, P.; Wei, W., Insights into the Enhanced Structural and Thermal Stabilities of Nb-Substituted Lithium-Rich Layered Oxide Cathodes. Acs Appl. Mater. Interfaces 2021, 13, (38), 45619-45629.
- Nie, L.; Wang, Z.; Zhao, X.; Chen, S.; He, Y.; Zhao, H.; Gao, T.; Zhang, Y.; Dong, L.; Kim, F.; Yu, Y.; Liu, W., Cation/Anion Codoped and Cobalt-Free Li-Rich Layered Cathode for High-Performance Li-Ion Batteries. Nano Lett. 2021, 21, (19), 8370-8377.
Point 2: Overall, it is not clear the scientific rationale for which the authors used many and different characterization methods without a strong correlation of the achieved results. For some aspects, the manuscript appears like a data and image mere gallery.
Response 2: Thank you very much for your valuable comments and your criticism, we have carefully revised the manuscript. Hope your concerns have been effectively improved.
Reviewer 2 Report
Questions/Comments Regarding Manuscript # 1597637
The writing of this paper is average. The manuscript contains some interesting achievements, and the results are well supported with sufficient discussions. However, some issues need to be seriously addressed before its acceptance and publication, as listed below:
- The literature cited is too old, so more articles in the past three years and some classical literatures should be cited. References should be updated.
- In addition, the authors should carefully elaborate the advantages of this study compared to the previous papers in the introduction section. The current introduction is not very much relevant to the significant achievements of current work. It contains numerous grammatical mistakes and needs severe corrections and additional discussions.
- Figure 4: Please include the Columbic efficiency variations in both the cyclic and rate tests.
- Please present dQ/dV curves for C/D profiles during cycling in Figure 4.
- Authors should clearly define all the acronyms and abbreviations when they first appeared, and consistently use them throughout the text.
- The manuscript needs a throughout proofreading as there are numerous language and grammar mistakes.
Author Response
Point 1: The literature cited is too old, so more articles in the past three years and some classical literatures should be cited. References should be updated.
Response 1: Thanks for your valuable comments and suggestion. The references have been updated in this revision.
Point 2: In addition, the authors should carefully elaborate the advantages of this study compared to the previous papers in the introduction section. The current introduction is not very much relevant to the significant achievements of current work. It contains numerous grammatical mistakes and needs severe corrections and additional discussions.
Response 2: Thanks for your valuable comments and suggestion. We have revised the introduction and highlighted all changes in this revision.
Point 3: Figure 4: Please include the Columbic efficiency variations in both the cyclic and rate tests.
Response 3: Thank you very much for your comments, the Coulumbic efficiency variations have been added in Fig 4a/4b.
Point 4: Please present dQ/dV curves for C/D profiles during cycling in Figure 4.
Response 4: Thank you very much for your comments. The dQ/dV curves have been provided Fig. S1.
Point 5: Authors should clearly define all the acronyms and abbreviations when they first appeared, and consistently use them throughout the text.
Response 5: Thanks for your valuable comments and suggestion. We have checked all acronyms and abbreviations carefully and corrected all errors.
Point 6: The manuscript needs a throughout proofreading as there are numerous language and grammar mistakes.
Response 6: Thanks for your valuable comments and suggestion. We have revised the manuscript throughout and corrected all language and grammar mistakes.
Round 2
Reviewer 1 Report
The authors have done some revision work, but it is still not enough to make the revised manuscript worthy of publication. The persistent lack of structural information acquired directly from HRTEM is a major weakness of the revised manuscript. I very much appreciated the authors' sincere statement (... it is really a flaw not to take corresponding DE schemes at the same time ...) but the problem remains. One possible solution is to include an ex-post FFT analysis of HRTEM images. After that the manuscript, with this additional information, will undoubtedly be eligible for publication.
Author Response
Referee’s comments:
The authors have done some revision work, but it is still not enough to make the revised manuscript worthy of publication. The persistent lack of structural information acquired directly from HRTEM is a major weakness of the revised manuscript. I very much appreciated the authors' sincere statement (... it is really a flaw not to take corresponding DE schemes at the same time ...) but the problem remains. One possible solution is to include an ex-post FFT analysis of HRTEM images. After that the manuscript, with this additional information, will undoubtedly be eligible for publication.
Response 1:
Thank you very much for your valuable and professional comments.
- We have revised the manuscript throughout again, especially, we have carefully rewritten the preparation section;
- Following your advice, we have conducted HRTEM analysis again and conducted ED and EDS mapping analysis simultaneously. EDS mapping and ED results of the LMR-Mo sample have been supplemented in Figure **, proving the homogeneous distribution of Ni, Mn, Co, and Mo elements on the LMR-Mo sample.